# Comparison of Chest Compression Quality Using Wing Boards versus Walking Next to a Moving Stretcher: A Randomized Crossover Simulation Study

**DOI:** 10.3390/jcm9051584

**Published:** 2020-05-23

**Authors:** Yukako Nakashima, Takeji Saitoh, Hideki Yasui, Masahide Ueno, Kensuke Hotta, Takashi Ogawa, Yoshiaki Takahashi, Yuichiro Maekawa, Atsuto Yoshino

**Affiliations:** 1Department of Emergency and Disaster Medicine, Hamamatsu University School of Medicine, Hamamatsu 431-3125, Japan; 41239258@hama-med.ac.jp (Y.N.); yasui@hama-med.ac.jp (H.Y.); 07485537@hama-med.ac.jp (M.U.); horitajanai0311@gmail.com (K.H.); 07485501@hama-med.ac.jp (T.O.); yoshitak@hama-med.ac.jp (Y.T.); yoshino@hama-med.ac.jp (A.Y.); 2Department of Cardiology, Hamamatsu University School of Medicine, Hamamatsu 431-3125, Japan; ymaekawa@hama-med.ac.jp

**Keywords:** chest compression, moving stretcher, wing

## Abstract

Background: When a rescuer walks alongside a stretcher and compresses the patient’s chest, the rescuer produces low-quality chest compressions. We hypothesized that a stretcher equipped with wing boards allows for better chest compressions than the conventional method. Methods: In this prospective, randomized, crossover study, we enrolled 45 medical workers and students. They performed hands-on chest compressions to a mannequin on a moving stretcher, while either walking (the walk method) or riding on wings attached to the stretcher (the wing method). The depths of the chest compressions were recorded. The participants’ vital signs were measured before and after the trials. Results: The average compression depth during the wing method (5.40 ± 0.50 cm) was greater than during the walk method (4.85 ± 0.80 cm; *p* < 0.01). The average compression rates during the two minutes were 215 ± 8 and 217 ± 5 compressions in the walk and wing methods, respectively (*p* = ns). Changes in blood pressure (14 ± 11 vs. 22 ± 14 mmHg), heart rate (32 ± 13 vs. 58 ± 20 bpm), and modified Borg scale (4 (interquartile range: 2–4) vs. 6 (5–7)) were significantly lower in the wing method cohort compared to the walking cohort (*p* < 0.01). The rescuer’s size and physique were positively correlated with the chest compression depth during the walk method; however, we found no significant correlation in the wing method. Conclusions: Chest compressions performed on the stretcher while moving using the wing method can produce high-quality chest compressions, especially for rescuers with a smaller size and physique.

## 1. Introduction

Correctly performed chest compressions exert significant survival benefits [1,2]. A depth of 5–6 cm is recommended in the international guideline to achieve high-quality compressions [3]. Rescuers can perform effective chest compressions while stationary; however, the efficacy of the compressions decreases with a moving stretcher [4,5]. When a rescuer walks alongside the stretcher and compresses a patient’s chest, the rescuer might experience poor coordination between their upper and lower body parts. Improvement of this issue is challenging unless specific training is added to conventional basic life support courses.

A novel solution is necessary to prevent improper chest compressions on a moving stretcher. In this study, we propose a simple method for moving chest compressions. This method involves the use of two wooden wing boards fixed to the stretcher’s frame with L-shaped brackets and C-clamps. Rescuers can board the wings and move with the stretcher instead of walking alongside the stretcher, allowing them to concentrate on the chest compressions.

Our study aim was to compare mobile hands-on chest compressions while walking (the walk method) and on wings (the wing method, Figure 1) using a mannequin. The following factors were investigated: (1) the quality of the chest compressions, (2) changes in vital signs and the degree of fatigue after the trials, and (3) the relationship between the quality of the chest compressions and the rescuer’s body size.

## 2. Methods

### 2.1. Study Design and Participants

The study was a prospective, randomized, crossover study, conducted between 4 June and 31 October, 2019. We called for participants who had completed and were found competent in the Basic Life Support course from the American Heart Association and/or Immediate Cardiac Life Support course from the Japanese Society for Emergency Medicine in Hamamatsu University Hospital. We excluded participants who were unable to perform enough chest compressions with a depth of over 5 cm to the mannequin on the stretcher while stationary. The participants were single-blinded to the study design, aim, and endpoints, and the investigators and assessors were not blinded. The study was conducted in accordance with the Declaration of Helsinki, the protocol was approved by the Hamamatsu University Ethics Committee (reference: 19–61), and written informed consent was obtained from all participants who volunteered for this study. The protocol was uploaded to the UMIN system. (#UMIN: 000036023). As a pilot study, 10 medical staff performed both the walk method and the wing method following the protocol presented in this study. The mean difference and standard deviation in compression depth were 5.5 and 12.5 mm, respectively. The required sample size was calculated to be 43 cases (α = 0.05, power = 0.8). Thus, we planned to prospectively enroll 45 healthy medical workers and students.

### 2.2. Study Protocol and Methods of Measurement

We measured or calculated the participants’ height, weight, body surface area, sole size, and limb length. The lengths of the participant’s arms and legs were measured from the humeral head to the palm and from the greater trochanter to the heel, respectively. A cardiopulmonary resuscitation (CPR) mannequin (Little Anne^®^; Laerdal Medical AS, Stavanger, Norway) was placed on a solid steel stretcher (model number KK-728B, Paramount Ltd., Tokyo, Japan) without a mat. The device used to evaluate the quality of the chest compressions (Shinnosuke-Kun^®^, Sumitomo Riko Ltd., Nagoya, Japan) was set up on the central chest of the mannequin according to the manufacturer’s guidelines [3]. The monitoring device was shielded from the rescuers during the trials. Before the trials, the rescuers reconfirmed the proper form for hands-on chest compressions, i.e., the positioning of their hands on the lower half of the sternum, maintaining a chest compression rate of 100–120 compressions/min and a chest compression depth of approximately 5–6 cm, and ensuring proper chest wall recoil [3]. Then, they practiced chest compressions a few times in preparation. The rescuers were able to listen to a metronome to the rhythm of 110 beats per minute to achieve the compression rate (100–120 compressions/min) during the trials [6].

In the wing method, we prepared two wooden wings composed of cedar (40 × 90 × 900 mm). Each wing was equipped with two L-shaped brackets (BS-568 NO68, Wakisangyo Ltd., Osaka, Japan) fixed by two screws (4 × 25 mm, YAHATA NEJI Co., Kitanagoya, Japan). They were secured to the frame of the stretcher using two C-clamps (GISUKE C-clamp 100 mm, Takagi Ltd., Sanjo, Japan; Figure 1). We instructed the participants to stand on the wooden wings, and the height of the stretcher was adjusted to the height of their knees. In the walk method, we instructed them to stand next to the stretcher, and adjusted the height of the stretcher to match the height of their knees. For both methods, a researcher pushed the stretcher at a speed of 1 m/s (3.6 km/h) from the starting point. The route was a 120 m long U-shaped flat corridor, requiring the researcher to turn right twice, turn around at a turning point, turn left two times, and return to the starting point (Figure 2). We set the CPR duration to two minutes, as that is the limit for effective continuous chest compressions before a decrease in compression quality occurs due to fatigue [3]. The order of the trials was randomized using the envelope method, with consecutive numbers following a randomization list; a note reading “walking first” or “wing first” was drawn. To allow the participants to recover, the second chest compression trial was performed at least 24 h after the first trial. After receiving the instructions, the blood pressure, heart rate, and respiratory rate of the participants were measured. The rescuers performed chest compressions for two minutes, then their vital signs were measured again. We used modified Borg scale scores to evaluate the subjective fatigue of the rescuers after CPR [6]. The primary endpoint was the depth of the chest compressions in the walk and wing methods. The secondary endpoints were the difference of the depth between men and women, changes in vital signs, the degree of fatigue after the trials, and the relationship between the quality of the chest compressions and the rescuer’s body size.

### 2.3. Statistical Analysis

The data recorded on the Shinnosuke-Kun device were imported into Microsoft Excel 2016 (Microsoft Corporation, Redmond, WA, USA) to analyze and categorize the compression rate and depth. Continuous data are expressed as mean ± standard deviation (SD) or medians with the inter-quartile range (IQR). The Kolmogorov–Smirnov test was used to evaluate the normal distribution. To analyze the data with a parametric distribution, a two-sided paired *t*-test was used to compare two continuous variables. Wilcoxon-signed rank test or Mann–Whitney U test was used to measure the non-parametric distribution, with or without paired variables. To analyze the relationship between the body size of the rescuers (height, body weight, body surface area, sole size, and length of the limbs) and the parameters of the chest compressions, we used the Spearman correlation coefficient test. *p*-values < 0.05 were considered statistically significant. All statistical analyses were performed using SPSS version 26 (SPSS Inc., Chicago, IL, USA) and EZR (Jichi Medical University, Shimotsuke, Japan).

## 3. Results

### 3.1. Participants

We enrolled 45 participants, consisting of 14 medical doctors, 16 nurses, and 15 medical doctors. All applicants were able to perform enough chest compressions; thus, no applicants were excluded. The median age of the participants was 26 years old (IQR: 23–31 years). There were 22 men and 23 women. Their median height, body weight, body surface area, arm length, leg length, and sole size were 164 cm (159–173 cm), 60 kg (50–66 kg), 1.66 m^2^ (1.47–1.82 m^2^), 51.5 cm (50–55.7 cm), 86 cm (82–91 cm), and 24.7 cm (23.5–26.8 cm), respectively. All participants completed the trials without any adverse events occurring.

### 3.2. Quality of the Chest Compressions

The average compression rates during the two minutes were 215 ± 8 and 217 ± 5 compressions in the walk and wing methods, respectively (*p* = ns). The average compression depth in the wing method (5.40 ± 0.50 cm) was greater than in the walk method (4.85 ± 0.80 cm; *p* < 0.01; Figure 3). The average chest compression depths in male participants were 5.3 cm (4.8–5.9 cm) and 5.4 cm (5.3–5.7 cm) in the walk and wing methods, respectively (*p* = ns). The average chest compression depths in female participants were 4.1 cm (3.8–4.6 cm) and 5.3 cm (4.9–5.7 cm), respectively (*p* < 0.01). The wing method achieved the target compression depth (5–6 cm) more often than the walking method (48.8% (37.2%–60.7%) and 24.6% (13.6%–37.5%) respectively; *p* < 0.01; Figure 4). The percent of chest compressions below the target (<5 cm) performed during the wing method (25% (15.5%–44.3%)) was less than during the walk method (59.4% (23.7%–83.4%); *p* < 0.01). The percent of chest compressions above the target (>6 cm) performed during the wing method (17.2% (6.9%–33.1%)) was not significantly different from the walk method (8.4% (1.88%–30.6%); *p* = ns).

### 3.3. Changes in Vital Signs

We observed no differences in the rescuers’ vital signs before the trials between the groups at the beginning of the walk or wing method (blood pressure: 120 ± 12 vs. 117 ± 13 mmHg, pulse: 73 ± 10 vs. 71 ± 12 bpm, and respiratory rate: 12 ± 3 vs. 12 ± 3 breaths/min, respectively; *p* = ns). Changes in blood pressure (14 ± 11 vs. 22 ± 14 mmHg), heart rate (32 ± 13 vs. 58 ± 20 bpm), and modified Borg scale (4 (interquartile range: 2–4) vs. 6 (5–7)) were significantly lower in the wing method cohort compared to the walking cohort (*p* < 0.01; Table 1). The blood pressure, pulse, and respiratory rate values after the trials were significantly greater than the values collected before the trials in both the walk and wing methods.

### 3.4. Relationship between Chest Compressions Quality and Rescuer’s Body Size

In the walk method, some of the rescuers’ size measurements, including height, body weight, body surface area, leg length, and sole size, were positively correlated with chest compression depth (Table 2); these values were also positively associated with the percentage of ideal chest compression depth. For the wing method, the only relationship observed between the rescuers’ size and the quality of chest compressions was a positive correlation between leg length and compression depth.

## 4. Discussion

High-quality CPR can improve the success rate of defibrillation and the return to spontaneous circulation (ROSC) [7,8,9]. The depth and frequency of chest compressions do not differ between kneeling and standing positions when performed on patients fixed in place [10]. The performance of chest compressions can be compromised due to the motion caused when transferring patients in cardiopulmonary arrest by stretchers. In this study, we showed that rescuers performed chest compressions on a mannequin poorly when required to walk alongside a moving stretcher. If a patient is transported on a stretcher over a long distance and requires resuscitation, the probability of ROSC is often low. Mechanical chest compressions would be the most effective and safe method of compression in this setting; however, it is not usually available as it requires time to set up and start the machine, and the battery may be dead or die during compressions. In these situations, performing conventional chest compression is necessary.

To improve the CPR success rate, various procedures have been developed. Straddling external chest compression is another method of performing compressions while a stretcher is moving with a patient [5]. The quality of the straddling chest compression method is as valid as conventional chest compressions from the side of the patient, yet, approximately 10.3 s are required for the rescuer to mount the stretcher and start the chest compressions [11]. The rescuer has to dismount the stretcher once they are fatigued and change positions with the next rescuer, which should be timely. This can be unsafe for the rescuer, as they may be so fatigued after the chest compressions that they might fall from the stretcher when dismounting. Interrupting chest compressions can adversely affect hemodynamics during CPR; therefore, the straddling method is not ideal [12]. The wing method allows for a prompt change between rescuers; they can stand on either side and switch immediately. However, if a rescuer performs longer durations of CPR, the rescuer could be still at risk of fatigue-related injury while on the wing boards, such as falling off while transferring a patient. Thus, if rescuers feel at-risk, especially in corners, they should adjust the height of the stretcher appropriately. Over-the-head chest compression is another alternative to conventional chest compression when a stretcher is moving. Some studies have described this possibility [13,14]; however, as with the straddling method, changing rescuers is difficult. This method may expose the rescuer to vomit and bodily fluids from the patient. To deal with various situations, it is important to prepare multiple methods for chest compressions.

The quality of the chest compressions was positively related to the body size of the rescuers in the walk method. This meant that rescuers with smaller bodies were less likely to perform strong chest compressions while walking, compared with those with larger bodies. As rescuers of smaller builds have to put more effort into achieving proper chest compressions, they may pay less attention to walking and stumble, leading to adverse events for both the rescuers and patients. Ideal chest compressions should be high quality for the patient, and of low physical effort and risk for the rescuers. To date, most researchers focused on the quality of chest compression alone; however, the fatigue of rescuers is also important, because it can affect the quality of the CPR. We showed that the walk method produces lower quality chest compressions than the wing method and increases the fatigue of the rescuers, especially those with smaller bodies. Thus, the wing method is favorable for rescuers with a small physique.

In this study, we moved the stretcher along a U-shape corridor to closely mimic the real transfer and movement of patients. We chose a round-trip course to allow rescuers to test both sides of the stretcher and to reduce any bias. In reality, various types of corners are found in hallways during patient transfer. Even when going around the corners in this research, all participants were able to maintain a stable body position and continue their chest compressions during the wing method, including the heaviest participant (weight, 88 kg; height, 173 cm). As they turned, they leaned over the mannequin, shifting their weight to the center of the stretcher; this posture prevented the stretcher from rolling over. However, the participants were shorter and lighter than the average provider in the Western world, so it is unknown if the same benefits would be obtained if the average providers were taller and bigger. We also need to produce wing boards that are compatible with various types of stretchers, rescuers, and conditions.

Before delivering a patient lying on a stretcher to a treatment room, the only preparation required to use the wing method is to fix two wings onto the stretcher and fasten them using C-clamps. These could be attached while a patient is boarded on the stretcher. If needed, the wings can be permanently attached to a stretcher. This procedure might also be useful for ambulance crews. When crews transfer the patient from where the patient was picked up to the ambulance, and from the ambulance to a bed in an emergency room, they seem to be unable to perform proper chest compressions. If their stretcher is equipped with wings, proper compressions can be achieved. The wings could also be implemented in the delivery of patients by helicopters.

In the future, the wings should be illuminated so others will recognize them as the stretcher passes by, helping this feature gain attention. An arrangement of folding wings could be safe and convenient. Bilateral skateboard-like wings may be more stable when rescuers ride on them and perform chest compressions.

## 5. Conclusions

The wing method allowed for better quality chest compressions and lower physical effort for the rescuers than the walk method during simulated cardiopulmonary resuscitation.

## 6. Limitations

In this study, the chest compressions were performed for two minutes. However, we need to examine future research avenues for chest compressions performed for different lengths of time, e.g., 30 s or three minutes. Secondly, we did not perform the chest compressions on a human body during this study. To ensure the validity of the present study, we need to apply this novel method to real patients in the future. However, the results of simulation-based studies can serve as a beneficial alternative for research questions that are difficult to answer in real cases because CPR performance shows a high agreement between real and simulated cases [15]. Third, we did not evaluate the ease of transition on and off the wing boards, the time it took for the transition of rescuers, or the time needed to assemble the wing boards. Fourth, even in the wing board cohort, the target depth was met in less than 50% of compressions. Thus, we need a more advanced CPR course, especially in terms of delivery for patients.

## Figures and Tables

**Figure 1 jcm-09-01584-f001:**
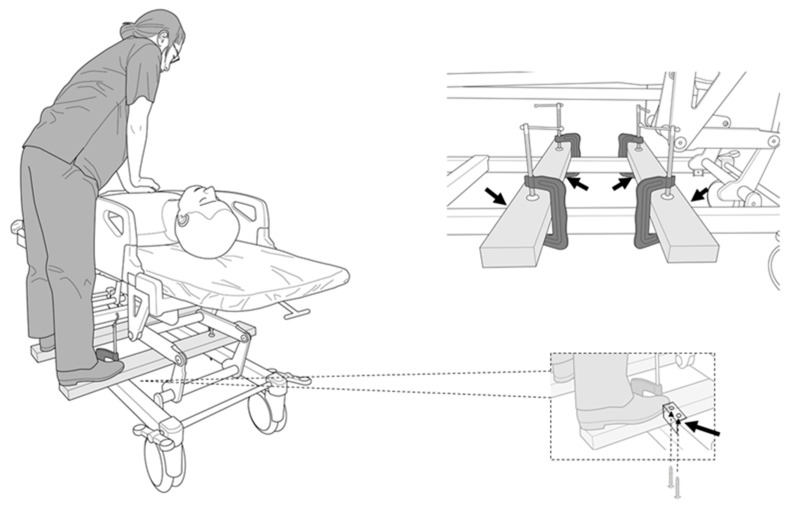
Chest compressions performed with wings attached to the stretcher (the wing method). The picture surrounded by the dotted line shows the L-shaped brackets screwed to the wings. The black arrows indicate the placement of the L-shaped brackets.

**Figure 2 jcm-09-01584-f002:**
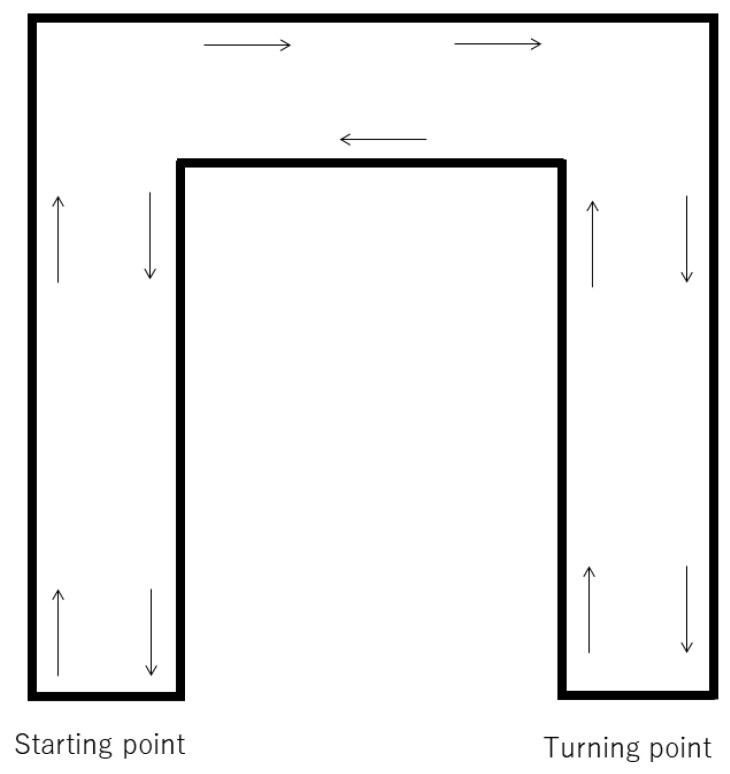
Outline of the U-shaped corridor used for the study’s delivery route. Each distance between the starting point and the first corner, the first and the second corner, and the second corner to the turning point was equal to 20 m. The total round-trip circuit was 120 m. The surface of the floor was rigid vinyl.

**Figure 3 jcm-09-01584-f003:**
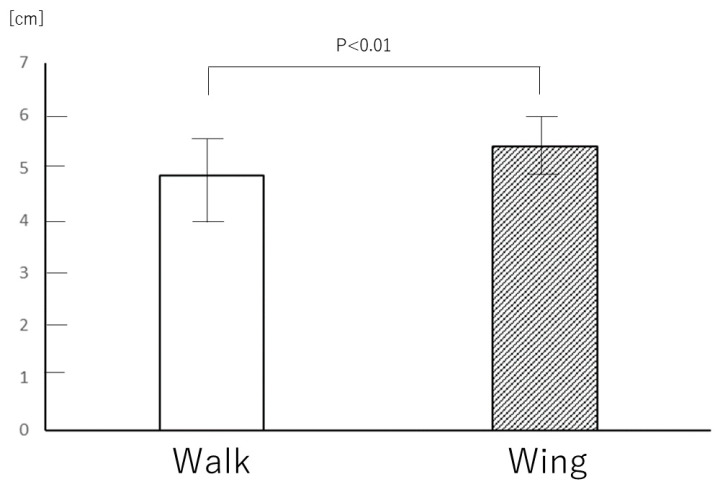
The difference between the average chest compression depths achieved in the walk and wing methods (5.40 ± 0.50 cm vs. 4.85 ± 0.80 cm; *p* < 0.01). The white bar represents the walk method and the grey bar represents the wing method. The data were analyzed by paired t-test.

**Figure 4 jcm-09-01584-f004:**
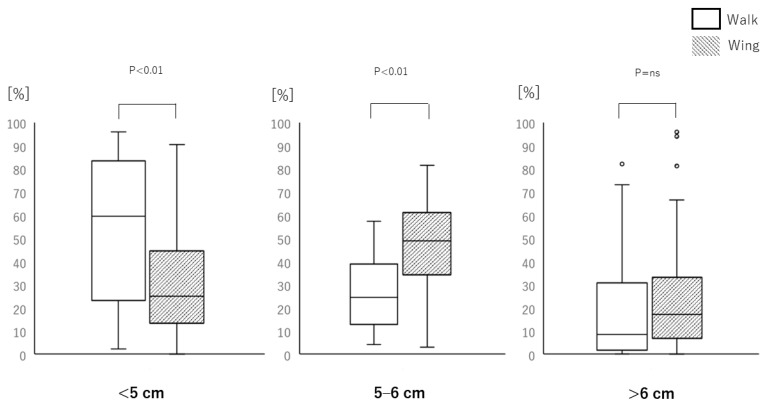
Data categorizing the percent of chest compression depths (<5 cm, 5–6 cm, or >6 cm) achieved with the walk (white bar) and wing (grey bar) methods. The central boxes represent values from the lower to upper quartile (25th to 75th percentile) within each group. The middle horizontal lines represent the median (50th percentile), and the whiskers represent the 5th and 95th percentiles. The white bars represent the walk method and the grey bars represent the wing method. The *p*-values above each category show whether there was a significant difference between the methods. The data were analyzed by Wilcoxon-signed rank test.

**Table 1 jcm-09-01584-t001:** Changes in vital signs and symptoms before and after the walk and wing methods.

Parameter	Walk	Wing	*p*-value
ΔBlood pressure (mmHg)	22 ± 14	14 ± 11	<0.01
ΔPulse (bpm)	58 ± 20	32 ± 13	<0.01
ΔRespiratory rate (breaths/min)	19 ± 4	12 ± 6	<0.01
Modified Borg scale	6 (5–7)	4 (2–4)	<0.01

Note: The data are presented as a mean ± standard deviation or median (interquartile range). The data in Δblood pressure, Δpulse, and Δrespiratory rate were analyzed by paired t-test; the data in modified Borg scale were analyzed by Wilcoxon-signed rank test.

**Table 2 jcm-09-01584-t002:** Correlation coefficients between the rescuer’s size and chest compression quality, considering chest compression depth and percentage of ideal chest compression depth.

Parameter	Depth		Percentage of Ideal Chest Compression Depth	
	Walk	Wing	Walk	Wing
Age	−0.08	−0.044	0.031	0.010
Height	0.622 **	0.240	0.584 **	0.283
Body weight	0.390 **	0.099	0.379	0.270
Body surface area	0.452 **	0.134	0.442 **	0.273
Arm length	0.502 **	0.265	0.520 **	0.086
Leg length	0.464 **	0.378 *	0.340*	0.110
Sole size	0.518 **	0.270	0.590 **	0.266

* *p* < 0.05. ** *p* < 0.01. The data were analyzed by Spearman correlation coefficient test.

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
