# Peer review of "Comparison of Chest Compression Quality Using Wing Boards versus Walking Next to a Moving Stretcher: A Randomized Crossover Simulation Study"

_jcm, 2020, doi:10.3390/jcm9051584_

Round 1

Reviewer 1 Report

This randomized crossover simulation study investigates chest compression quality when performing chest compressions walking next to a moving stretcher vs. standing on a “wing” on the stretcher. While it is known that quality of chest compressions when moving a stretcher is substandard to guideline recommendations, some studies have already found that other alternatives such as straddling the patient may be superior to walking next to the stretcher (Lei, 2010: doi:10.1016/j.resuscitation.2010.05.017; Shinchi 2019: doi: 10.1371/journal.pone.0216739). Accordingly, this study would have been stronger if comparing to e.g. the straddling position when doing chest compressions. The novel thing in this study is to show that chest compressions can be improved by adding a simple “wing” compared to walking next to the stretcher. It also shows that the effect depends on how tall the chest compressors are.

The study is overall well-written but some descriptions are missing in the methods section on e.g. randomization and blinding. A few elements in the discussion and conclusion are a bit speculative and the discussion could be strengthened by discussing this study a bit more in relation to comparable studies. Some specific comments:

Introduction:

First line: Reference 1 is an expert statement – please include original literature.

Lines 44-48: When stating the aim of the study – please specify the primary endpoint in the aim – i.e. to compare the effect on chest compression depth.

Methods:

Please specify in- and exclusion criteria for participants.

line 69: It is mentioned that “rescuers reconfirmed the proper form for hands-on chest compressions”. However, it is unclear what this entails? Did the rescuers practice on a manikin until proper hand position, rate, depth, and chest recoil was obtained? How many compressions were needed to be correct before moving on to the trial?

line 85: Please elaborate on randomization in the study. Did you use envelopes with consecutive numbers following a randomization list? Or did they draw from a pile? Were the envelopes opaque?

Moreover, please elaborate on blinding: Were participants blinding to study design, aim, and endpoints? Were researchers/ assessors blinded?

Please describe the prospectively defined primary- and secondary endpoints in the methods section.

Lines 105-106: It is described which statistical test that was used for all endpoints except from chest compressions. Please describe which test was used to compare groups for chest compressions.

Results:

If available, please provide additional demographic data on participants (perhaps in a table): What was the profession, training, experience from clinical resuscitation attempts, time since last CPR training etc?

Lines 120-121: Chest compression rates of 215 vs. 217/min seem obscure – do the authors mean 115 vs. 117/ min?

Do the authors have data on chest compression recoil or/ and release velocity?

Please include difference with 95% confidence interval whenever possible as it is more informative than the p-value alone.

Lines 136-140: This text belongs in a figure legend rather than in the main text.

Correlation between body metrics and depth – method?

The data shows that standing on the wing is superior to walking next to the stretcher when performing chest compression for 120 seconds. However, is time an important factor like body size? Did the authors also consider at time? This may have clinical implications if the clinician has a very short time (e.g. 20-30 sec) to move the patient in- or out of the ambulance vs if the clinician needs to move the patient over a longer period. If not, this may be added to the discussion/ limitations.

Discussion:

Line 162-163: The authors provide 3 references for the statement that chest compressions >5cm is associated with improved survival. However, 2 of the references show that increasing depth is associated with improved survival (not 5cm specifically). Accordingly, I suggest rephrasing to state that increasing depth is associated with improved survival. Edelson et al shows that performance debriefing improves depth and survival (not specifically that depth improves survival).

Line 175: Please provide a reference

Lines 188-190: It seems a little speculative as I am not aware of any evidence showing that it will create safety issues. Therefore, I suggest omitting or rephrasing.

Lines 197-198: From which side were the chest compressors standing? Always on the same side or on different sides? There is a potential big difference between how difficult it is to do compressions in a turn depending on which side you are standing.

When transporting the patient, it can be considered both to walk, to straddle, to do over-the-head CPR, or to kneel next to the patient (Chi Resuscitation. 2008 Jan;76(1):69-75). Recognizing that the last thing may not always be feasible when transporting on a stretcher, it still deserves mention.

The authors mention the study by Shinchi et al, 2019. Another study looking at this is Lei, 2010: doi:10.1016/j.resuscitation.2010.05.017. I think it deserves mention that Shinchi et al. support the notion of walking being an inferior method when doing compressions. Also, it should be noted how this study differ from Shinchi et al. and Lei et al. and how this study therefore adds to the literature.   

The invention of a “wing” perhaps deserves a little more discussion. Is it feasible to implement and does it affect the space in e.g. an ambulance or a helicopter?

Conclusions: This is a simulation study – accordingly it should state during simulated cardiopulmonary resuscitation instead of “during delivery of a patient”. The last sentence is rather speculative and does not belong in the conclusion.

Limitations: This was performed with turning to only one side and a fixed speed – i.e. a very controlled environment. This is an important limitation. Moreover, the participants were shorter and lighter than the average provider in America or Europe. It is unknown if the same difference would apply when the average provider is a lot taller and bigger.

Author Response

We wish to express our appreciation to the reviewer for honest comments, which have helped us improve our paper. We would like to explain what is new and valuable in this paper in the file attached. 

Reviewer 2 Report

I think that it might have some novelty in the situation which the rescuer compressed the chest on moving stretcher but it is a narrow aspect of high-quality CPR.

There are Several major problems

#1. Please describe the manuscript according to the CONSORT check list

#2. I would suggest the title such as “Impact of Chest compressions while using wing boards on a moving stretcher: a randomized crossover simulation study”

#3. Please write the Clinical trials number (NCTxxxxxx)

#4. Methods

  1. Please, add the program, which method, how many participants, how much drop rate about the sample size.
  2. Please describe the inclusion and exclusion criteria about participants
  3. Was this study open-label study? single-blind?
  4. What was the primary outcomes and secondary outcomes?
  5. After receiving the instructions, the blood pressure, heart rate, and respiratory rate of the participants were measured. The rescuers performed chest compressions for 2 minutes; thereafter, their vital signs were measured again. We used the modified Borg scale scores [5] to evaluate the subjective fatigue of the rescuers after CPR. “
  • Please add the reference about the relationship between the fatigue and V/S.
  1. Statistics : This was randomized cross-over study. I would like to consult this to the experts.

“a two-sided paired t-test or Mann–Whitney U test was used to compare two continuous variables, with or without a normal distribution.”

: Which test was used to evaluate the normality?

It is proper to analyze Wilcoxon-sign rank test instead of Mann-whitney U test

  1. I think that the Sex will an important factor as the categorical variable to the chest compression and it should be included when analyzed.

#3. “The quality of the straddling chest compression method is as valid as conventional chest compressions from the side of the patient. Yet, about 10.3 seconds are required 175 until the rescuer can mount the stretcher and start the chest compressions. [10] “

  • How does it take to compress the chest using your wings?

Author Response

(The authors gave the same response as above.)

Reviewer 3 Report

The authors present a comparative analysis of a novel technology aimed at improving chest compression depth while performing compressions on a moving stretcher. It is a prospective, crossover study utilizing 45 “medical workers and students”. The authors intent to compare walking verses implementation of a wing board is appropriate for this purpose. The authors measured compressions rate and depth as the primary outcome metrics. Secondary metrics included individual body metrics in relation to chest compression quality and vital signs pre- and post-compressions as a measure of compression fatigue. The authors found a significant improvement in chest compression depth with implementation of the wing boards. However, both groups failed to obtain the target depth more than 50% of the time. On review of secondary analysis the authors found height and body weight improved compression depth in the walking group, and there was less vital sign change in the wing group compared to the walking group after performing chest compressions. While the authors did create a novel method for improving chest compression quality, which did improve compression quality and decrease fatigue, the authors failed to acknowledge several limitations to their study design including significant concerns in the ability to safely implement the wing board in real-life scenarios.

TITLE COMMENTS:

The authors should consider strengthening the title by including study design and primary metric studied.

Consider switching the title to: “Comparison of Chest Compression Quality Using Wing Boards Verses Walking next to a Moving Stretcher”.

ABSTRACT COMMENTS:

  1. Include the hypothesis in the background section.
  2. What are “medical workers”? This should be defined more clearly.
  3. Line 16 should include rate in addition to depth as a measured quality metric
  4. Grammatical suggestions:
    1. Line 13 - the rescuer performs low quality chest compressions, it is the patient who experiences low quality chest compressions
    2. Line 17/18 - clarify how analyzing vital signs has a relationship on “its quality and body size”. My understanding was the vital sign analysis was to compare compressor fatigue during walking verses wing board compressions and body size was to look at overall compression quality of the compressor.
    3. Line 20 - consider changing the line starting with “The increases in blood pressure…” to “Changes in blood pressure (14±11 mmHg vs. 22 ± 14 mmHg), heart rate (32 ± 13 bpm vs. 58 ± 20 bpm), and modified Borg scale (4 [Interquartile range: 2–4] vs. 6 [5–7])  were significantly lower in the wing method cohort compared to the walking cohort (p<0.001).  

MANUSCRIPT COMMENTS:

Introduction:

  1. Clarify the statement (Line 36) “the rescuer experiences poor coordination between their upper and lower body parts” and its impact on CPR quality.
    1. Consider including additional sources looking at the decline in CPR quality during transport.
  2. All but one of the resources was published before 2008. There have been significant advances in CPR quality monitoring and metric reporting since then. I feel they could enhance the introduction with updated literature.
  3. Gramatical considerations -
    1. Line 32 starting with “Rescuers can perform…” It is unnecessary to include all of the positions in which rescuers can perform effective chest compressions. Instead they could change it to “Rescuers can perform effective chest compressions while stationary, however, the efficacy of the compressions decreases with a moving stretcher”.

Materials and methods:

  1. Clarify “medical workers and students”.
  2. How did you determine competency in BLS?
  3. Include rate as a study metric on page 3 line 103 as it ins included as a comparison metric in the results section.
  4. Is there a reason you left out fitness/exercise tolerance in addition to body size metrics?
  5. The authors used a Pearson’s correlation coefficient for body size and quality of compressions however, the Pearson’s correlation coefficient would assume normal distribution. The use of body metrics and compression quality would likely suggest that the distribution is non-Gaussian.  In this scenario, Spearman correlation is likely the best test. 
  6. Well described study design and graphic representations of the wing board method.
  7. Gramatical considerations
    1. Consider reformatting paragraph 2.1 to start with the study design, sample size, dates, IRB information, and end with the pilot study/power.
    2. Consider moving “as volunteers” in line 58 on page 2 to line 53 “consent was obtained from all participants who volunteered for this study”.

Results:

  1. Consider making the results section more concise.
    1. Figures 3 and 4 have paragraphs below reiterating exactly what is in the table and written in the the first paragraph of section 3.2.
    2. Instead of writing out all of the pretrial vital signs on page 5 line 143-145 they could put them in a table.
  2. Grammatical considerations
    1. Page 4 line 123 starting with “The percent of ideal chest compressions…” could be reworded to “The Wing method was more likely to achieve target compression depth (5-6 cm) compared to the walking method (48.8% [37.2-60.7%] and 24.6% [13.6-37.5%] respectively; p<0.001).
    2. Consider using below and above target depth instead of the percent of insufficient and excessive chest compressions (Page 5 line 125 and 127 respectively)
    3. Consider changing the sentence begging with “After the trials, the increases in blood pressure… on Page 5 line 145 to "Changes in blood pressure (14±11 mmHg vs. 22 ± 14 mmHg), heart rate (32 ± 13 bpm vs. 58 ± 20 bpm), and modified Borg scale (4 [Interquartile range: 2–4] vs. 6 [5–7]) were significantly lower in the wing method cohort compared to the walking cohort (p<0.001) as previously stated in the abstract section.

Discussion:

  1. The authors made multiple assumptions with only one limitation to their study. However, I found multiple limitations in this study, many of which are similar reasons they used to support their method over alternative methods of CPR on a moving stretcher.
    1. The authors state their wing boards are superior to other methods of CPR on moving stretchers because it allows for decreased compression interruptions and smoother transitions, however the authors did not comment whether or not they tested the ease of transition on and off the wing boards, the time it took for the transition of compressors, or the time it took to assemble the wing boards.
      1. The study is run in 2 minute intervals to minimize fatigue, however, they study could have benefitted significantly by including longer durations of CPR to assess ease and duration of transitions between compressors. Also, it is not guaranteed compressors will switch every 2 minutes, especially in transport. The compressors could also fatigue on the wing boards and fall off in movement or during a transition and injure themselves.
      2. The study does not address how long it took to assemble the wing boards. It referenced the possibility of having the beds are already equipped with wing boards, A delay in assembling the wing boards could compromise CPR quality. It could also delay life saving procedures (as the initial intent for transporting these patients treatment rooms for advanced in-hospital procedures).
  1. The authors did not comment on their overall CPR quality compared to other studies and current recommendations. In both the walking and the wing board cohorts the target depth was met in less than 50% of compressions.
  2. The authors did not include the maximum weight the wing boards could hold. The median body weight was 60 kg, which is not generalizable worldwide.
  1. While this is a novel idea that did show improved CPR depth and decreased compressor fatigue, I am not sure convinced on the feasibility of wing boards with safety being my biggest concern.
    1. The heaviest compressor was 88 kg. Per the authors, they had to lean over the manikin, which shifting their weight to the center of the stretcher and prevented the stretcher from flipping over. This poses a significant risk for injury for both the patient and the compressor.
    2. The increased width of the wing boards (especially if they are set up on both sides of the bed) could injury bystanders as they are moving the patient to the treatment areas.
    3. If the wing boards were left on the beds (as mentioned as a possible intervention) it poses risk to bystanders with any movement of the stretcher. It may also compromise the integrity of the wing boards as they are likely to be hit on corners as the stretchers are routinely used. A compromised wing board could injury the compressor and cause a potential delay in resuming high quality CPR.
  2. “When the clues transfer the patient from where the patient was picked up to the ambulance and from the ambulance to a bed in an emergency room, they cannot often perform proper chest compressions.” This statement needs references. Also, multiple studies have shown improved EMS compression quality in transportation compared to at the scene with mechanical compressors and real time audiovisual feedback. However, the compression depth often fails to meet the recommendations as did the depth in this study, therefore, the statement “if their stretcher is equipped with the wings, proper compressions can be achieved.” would be misleading.
  3. Gramatical Considerations:
    1. Page 6 line 1: for ease it should read “High quality CPR can improve the success rate…” instead of “in CPR, chest compression of”.
    2. Page 6 line 164 reads “during the transferring patient with cardiopulmonary arrest can been compromised…” The authors should clarify transferring (I assume they mean moving stretcher and not transferring beds) as well as can been compromised.
    3. Page 7 line 207 I assume it is supposed to say ambulance crews, but has been typed clues.

Graphs/Tables/Figures

  1. All p values of “0.000” can be easily represented as “ <0.01”.

Author Response

We wish to express our appreciation to the reviewer for honest comments, which have helped us improve our paper. We would like to explain what is new and valuable in the paper in the file attached. 

Round 2

Reviewer 1 Report

Thank you for the revised manuscript and a thorough response to previous comments. The manuscript has been substantially approved. As an overall comment, the manuscript would benefit from a thorough English grammar check.

A few specific comments:

  • Abstract line 33: I do not believe that “seldom” (or seldomly as it is used as an adverb) is the correct word as you are discussing whether there is a correlation between rescuer size and compression depth or not. I think it would be more correct stating e.g. that there was a weak correlation or that there was no significant correlation.
  • Methods line 72: It is described that the authors include “14 medical doctors, 16 nurses, and 15 medical students”. As it is described in the methods section, the reader would speculate that the authors prospectively decided to include this exact distribution of medical doctors, nurses, and medical students. I am although not sure whether this is the intention by the authors? If not, I suggest writing e.g. that you included all doctors, nurses, and medical students employed at XX department at YY hospital and then specify the resulting distribution of participants in the results section.
  • Methods line 76: The authors included information stating that participants were single blinded. Do the authors mean that participants were blinded to the study design, aim, and endpoints but investigators and assessors were not blinded? Please describe clearly.
  • Results line 153: The sentence “All participants were not excluded” makes no sense. Do the authors mean that no participants were excluded? Please rephrase the sentence.
  • Line 156-157: The following sentence does still not make sense: “The average compression rate during the 2 minutes was 215±8 compressions/min and 217±5 compressions”. Do the authors mean that the average number of compressions during the 2 minutes was 215±8 compressions/min and 217±5 compressions?
  • Lines 159-161: Thank you for providing this very interesting information on sex differences. Please specify in the methods section whether this stratification on sexes was prospectively defined (i.e. defined in advance in the protocol) or whether this was a post-hoc analysis.
  • Figure 3: There is a “3A” and “3B” i.e. two different plots of bars. Please specifiy what the difference is between the two sets of bars in the figure legends.
  • Discussion lines 216-217: I do not believe there is any evidence to support that mechanical chest compressions are safer compared to walking next to a stretcher (unless we are discussing ambulance transport). However, mechanical chest compressions will provide a consistent compression quality during transport while walking next to the stretcher may not.
  • The authors use multiple statistical tests without adjustment for multiple testing and the study itself is not powered to show a difference in the proportion of compressions within guideline recommendations. Accordingly, the risk of a type 1 error deserves mention in the limitations.

Author Response

We wish to express our appreciation to the reviewer for honest comments, which have helped us improve our paper. We would like to explain what is new and valuable in this paper in the file attached. This paper has undergone English language editing by MDPI. 
